# Hesperidin: A Review on Extraction Methods, Stability and Biological Activities

**DOI:** 10.3390/nu14122387

**Published:** 2022-06-09

**Authors:** Krystyna Pyrzynska

**Affiliations:** Department of Chemistry, University of Warsaw, Pasteura 1, 02-093 Warsaw, Poland; kryspyrz@chem.uw.edu.pl

**Keywords:** hesperidin, extraction, determination, stability, biological activities

## Abstract

Hesperidin is a bioflavonoid occurring in high concentrations in citrus fruits. Its use has been associated with a great number of health benefits, including antioxidant, antibacterial, antimicrobial, anti-inflammatory and anticarcinogenic properties. The food industry uses large quantities of citrus fruit, especially for the production of juice. It results in the accumulation of huge amounts of by-products such as peels, seeds, cell and membrane residues, which are also a good source of hesperidin. Thus, its extraction from these by-products has attracted considerable scientific interest with aim to use as natural antioxidants. In this review, the extraction and determination methods for quantification of hesperidin in fruits and by-products are presented and discussed as well as its stability and biological activities.

## 1. Introduction

Hesperidin (3,5,7-trihydroxyflavanone 7-rhamnoglucoside, hesperetin-7-*O*-rutino- side) belongs to flavanone compounds, one of the flavonoids subclasses (Figure 1). It has been recently extensively evaluated for its health-promoting and pharmacological effects and is used in a treatment of type 2 diabetes, cancer and cardiovascular diseases, neurological and psychiatric disorders, as well as a radioprotector [1,2,3,4,5,6]. Administrations of hesperidin can also benefit a variety of cutaneous function in both normal and diseases skin [7].

Hesperidin and its derivatives are characteristic compounds of citrus fruits (*Rutaceae* family) such as orange (*Citrus sinensis*), grapefruit (*Citrus paradise*), tangerine (*Citrus reticulata)*, lime (*Citrus aurantifolia*) and lemon (*Citrus limon*). Their content in citrus fruits depends on fruit variety, part of the fruit itself, climate and degree of maturation. According to the review of Gattuso et al. [8], 100 mL of an appropriate juice contains 20–60 mg of hesperidin for orange, 8–46 mg for tangerines, 4–41 mg for lemon and 2–17 for grapefruit. Citrus flavedo (the coloured outer layer of the peel) and albedo (a white soft middle layer part) contain higher amounts of hesperidin in comparison to hand-squeezed juice [4,9]. Commercial juices, squeezed also with peel constituents, are rich in this flavanone. Apart from citrus fruits, hesperidin was also found in mint plants (*Mentha*) [10,11], honeybush (*Cyclopia mac ulata*) [12] and aromatized tea [13]. It is worth mentioning that hesperetin, the aglycone form of hesperidin, is less dominant in nature than its glycosides.

After oral intake, hesperidin is hydrolysed by the gut microbial rhamnosidases in the small intestine and mainly in the colon into aglycone form (hesperetin), and is then converted to glucuronides in the large intestine. Hesperetin appears in plasma 3 h after ingestion in the form of glucaronides (87%) and sulphaglucooronides (13%), reaching a maximum between 5 and 7 h [14]. They later undergo ring fission and were catabolized, producing phenolic acids and their respective metabolites. However, hesperidin exhibits low water solubility and limited bioavailability [1,6,15,16,17,18]. Various approaches such as the micronization and encapsulation of hesperidin have been proposed, particularly for drug productions, to improve its bioavailability, stability and controlled release [17,19,20,21].

In this review, the extraction and determination methods for quantification of hesperidin in different kinds of samples were presented and discussed as well as its stability and biological activities.

## 2. Extraction

Due to its biological activities, hesperidin is often used in the food, cosmetic and pharmaceutical industries. Optimal extraction processes from plant materials with high quality and purity are implied in these applications. Several procedures for the extraction of flavonoids, including hesperidin, have been explored, also taking into account those that are environmentally friendly [22,23,24,25,26,27,28,29,30,31,32,33,34,35,36]. Common extraction methods include dipping, percolation, reflux or continuous reflux. The quality of an extract and efficiency of a procedure are influenced by several factors such as solvent type, temperature, extraction time and liquid–solid ratio.

Methanol and ethanol, or their mixture with water at different proportions, as well as dimethyl sulfoxide (DMSO), are usually used. Maceration and Soxhlet extraction are increasingly being replaced by advanced techniques to increase efficiency and selectivity. They are generally faster, more environmentally friendly, and with higher automation levels. Several methodologies based on accelerated solvent extraction (ASE) [28], microwave-assisted extraction (MAE) [29,30], ultrasound-assisted extraction (USE) [23,29,30,31,32], subcritical water extraction (SWE) [33,34], pressurized liquid extraction (PLE) [35] and high hydrostatic pressure (HHP) [36] have been used for isolation of hesperidin from plant materials. Application of mathematical and statistical methods to the analysis of chemical data, like experimental design, response surface analysis and principal component analysis, have been often used for determining the optimum extraction conditions [23,27,28,29,30].

Two sequential extractions using 90% methanol for 20 min agitation at 55 °C were considered as optimal for isolation of hesperidin (24.77 mg/g dw) from sweet orange pulp (*Citrus sinensis* L.), while for 90% ethanol under the same conditions 17.9 mg/g of hesperidin was obtained (with significant differences at *p* < 0.05) [27]. Gómez-Mejia et al. [23] proposed extraction from orange peels with low concentration of ethanol (maximum 40%, *v*/*v*) run in an ultrasound bath for 10–15 min. It was stated that the use of a relatively high temperature (90 °C) could be compensated for by a very rapid and efficient extraction process. On the other side, a DMSO:methanol mixture (1:1) turned out to be a better medium during 10 min of ultrasound operation for the extraction of hesperidin from mandarin (*Citrus reticulata* Blanco) rinds [31]. The yield of hesperidin from peels of *Citrus unshiu* fruits after the MAE process (70% aqueous ethanol, heating 140 °C, 7 min) was comparable to the amount extracted at room temperature for 30 min using the DMSO:methanol (1:1) mixture [29]. The maximum yield of hesperidin from *Citrus unshiu* peel using SWE method was obtained at 160 °C for an extraction time of only 10 min [33]. It was 1.9-, 3.2- and 34.2-fold higher than those when 70% ethanol or methanol and hot water, respectively, were used.

Recently, room temperature ionic liquids (ILs) and deep eutectic solvents (DESs) were introduced as a new kind of alternative solvents for extraction and/or purification of bioactive compounds [37,38,39,40]. ILs are organic salts consisting entirely of ions, relatively bulky organic cations (imidazolium, pyridinium) with different tailorable characteristics, and small inorganic anions (Cl^−^, Br^−^, BF_4_^−^, PF_6_^−^). They show a melting point generally below 100 °C. Several imidazolium-based ionic liquids with different alkyl positions and alkyl chain lengths were evaluated in the microwave-assisted extraction of flavonoids from plant material [41]. The best efficiency of extraction (temperature of 80 °C, time of 60 min, microwave power of 300 W and IL concentration of 1.0 mol/L) was obtained for 1,3-dibutyl-2-methyl imidazolium bromide. Tang et al. [42] established optimal extraction parameters for isolation of some flavonoids from plant leaves using 1-decyl-3-methylimidazolium bromide as additive (2.5 mg/mL) in 200 mL of methanol in Soxhlet extraction at 200 °C for 8 h. 1-Hexyl-3-methylimidazolium tetrafluoborate was used for the separation of hesperidin, hyperoside and rutin in vacuum microwave-assisted extraction [43].

Deep eutectic solvents, formed from Lewis or Brönsted acids and bases, exhibit physicochemical properties very similar to ILs, such as negligible volatility, high thermal and chemical stabilities, but are less toxic and more biodegradable [44]. They can be easily obtained by mixing at proper ratios solid compounds, after mild heating. The resulting eutectic mixture has a melting point much lower than that of the individual components. In a study by Bajkacz and Adamek, 17 different natural DES systems with 2 or 3 components based on choline chloride, acetylcholine chloride, choline tartrate, betaine and carnitine with different compositions were evaluated for the extraction efficiency of flavonoids, including hesperidin, from fruits, vegetables and spices [45]. The best extraction yield of the target compounds (>70%) was reached using a 30% water solution of acetylcholine chloride/lactic acid (molar ratio 2:1) and 30 min extraction time at 60 °C. The relationship between the extraction yields of flavonoids from citrus peels and physicochemical properties of the DESs was examined by Xu et al. [45]. Choline chloride (ChCl) and sugars, amides, alcohols and carboxylic acids as the second compound were used at their molar ratio 1:2. For amide- and carboxylic acids-based DESs, the efficiency of hesperidin extraction linearly increased in the order: ChCl–urea < ChCl–*N*-methylurea < ChCl–acetamide, which correlates with the same trend of increasing log K_O/W_ values.

Table 1 presents the recent examples of extraction conditions for the isolation of hesperidin from different plant materials.

## 3. Determination of Hesperidin

Several methods have been developed for the analysis of hesperidin alone and/or with other flavonoids in different kinds of samples. Among them, the most widely used are based on reversed-phase high-performance liquid chromatography (RP-HPLC) coupled to diode array detection (DAD) and/or mass spectrometry (MS) with methanol/acetonitrile/water gradient [46,47,48,49]. The addition of ionic liquid (didecyl dimethylammonium lactate) to methanol/water eluent was proposed for shortening the analysis time and improving the peak symmetry in the analysis of pharmaceutical formulations [50].

Besides liquid chromatography, thin layer chromatography [51] capillary electrophoresis [52] and electrochemical methods [53,54,55] were also applied for hesperidin determination in citrus juices and peels. In addition, electroanalytical sensors can also be used in miniaturized form with a high degree of portability.

Spectrophotometric methods for determination of hesperidin are fast, cost-effective and require less expensive equipment. Bennani et al. [56] used two spectrophotometric methods to simultaneously determine the diosmin and hesperidin content in their binary mixture. First, the derivative spectrophotometry was applied by the zero-crossing measurements based on the elaboration of the linear calibration graphs of first derivative values, plotted at 269 nm and 262.5 nm, respectively, for hesperidin and diosmin. Then, in the second method, based on the calculation of the peak absorbance ratio at λ_max_ of both compounds, the percentage of drugs was determined. Another spectrophotometric method was based on the formation of hesperidin complex with Zn(II) in 70% (*v*/*v*) methanol at pH 3.1 (λ_max_ 283 nm, logβ_2_ = 17.01) [57]. The developed method was applied for the quantification of hesperidin in tablets and several samples of orange juices.

## 4. Stability of Hesperidin

Hesperidin, as other flavonoids, during different extraction processes might degrade when exposed to light, air and elevated temperature. The presence of the oxidative enzymes and free radicals released during extraction could also promote a series of degradation reactions [29,35,58,59,60]. It was found that application of sonication caused degradation of hesperidin standard solution as well as other flavonoids [58]. The highest decomposition in the methanol solution was observed for myricetin (40% decrease), followed by hesperidin (30%). As a contrast, all studied compounds were stable during heating reflux in a water bath for 30 min and maceration for 24 h (recovery above 95%). Majumdar et al. reported that aqueous solutions of hesperidin (5 µg/mL) did not demonstrate any decrease in its content up to 2 months in the pH range of 1–7.4 at 25 and 40 °C [59]. The only exception was pH 9, where the degradation rate constants were 0.03 and 23 at 25 and 40 °C, respectively, most likely by alkaline hydrolysis.

The effect of hesperidin degradation depends also on the type of food matrix [60,61]. Biesaga et al. [61] presented a study concerning the stability of hesperidin and other flavonoids in some food samples (honey, apple and onion) during different extraction processes. After addition of 60% (*v*/*v*) aqueous acidified methanol (pH 2), the solutions were subjected to heating in a water bath for 15 min, sonication for 5 min (20 Hz) or microwave irradiation (90 W) for 1 min. The apple matrix seems to stabilise these compounds to a large extent, while the highest degradation was observed for the honey samples, particularly for heated reflux and MAE conditions. Low recovery (68%) of hesperidin from maize samples during USE extraction was reported, similar to its standard solution [59].

Zhang et al. investigated the effects of storage condition and heat treatment during pasteurization of juice from orange fruits (*Citrus sinensis* Osbeck cv. Newhall) on the hesperidin concentration [46]. They found that the hesperidin concentration decreased when the juice was stored at room temperature and even at 4 °C, but at a slower rate. After 6 and 20 h of storage at −18 °C, the hesperidin concentrations in the juice were 576 and 533 mg/L, respectively, compared to 658 mg/L determined in the freshly squeezed juice.

The concentration of hesperidin in juice subjected to heat treatment (80 °C for 10 min) was almost the same as that in freshly squeezed juice. However, when this heat-treated juice was kept at room temperature, its concentration reduced gradually from the initial value of 577 mg/L to 294 mg/L after 30 days of storage. Similar results were obtained for untreated juice. Additionally, it was confirmed that the reduction of hesperidin concentration was not caused by enzymatic degradation (heating at 100 °C for 10 min) [46]. It was suspected that the hesperidin had sedimented from the juice during storage. It was later confirmed after determination of the whole content of this compound (containing both soluble and precipitated hesperidin) using extraction with DMF. The authors assumed that the precipitation of hesperidin might be due to the gradual decomposition of vitamin C.

## 5. Biological Activities

The human benefits of hesperidin taken from fruits and beverages or pharmaceuticals depend mainly on its bioavailability. The bioavailable fraction is commonly defined as the quantity of a given substance released from food that is absorbed through the intestinal barrier and enters the blood stream, reaching the systematic circulation, which is then distributed to organs and tissues and is transformed into a biochemically active form, which is effectively used by the organism [62]. The bioavailability of hesperidin is low due to its low aqueous solubility, absorption, and modification by microorganisms in the gastrointestinal tract and rapid excretion [16,18,63]. It is considered that the limiting step of the hydrolysis and absorption of hesperidin is the enzymatic activity α-rhamnosidase, which takes part in these processes. Although hesperidin is poorly absorbed and rapidly eliminated, it has a reasonable half-life of 6 h [64].

The biological and pharmacological properties of hesperidin have been extensively studied to reveal its antioxidant, anti-inflammatory, anticancer, antiviral effects, protective cardiovascular disorders and neurodegenerative properties, among others. Examples of the main biological activities of hesperidin are presented in Table 2.

A number of researchers have examined the antioxidant activity of hesperidin using various assays [65,66,67,68,69]. Its antioxidant properties are expressed mainly by direct free radical scavenging or indirectly by inhibition of prooxidative enzymes that participate in the generation of these radicals as well as by chelation of transition metals which participate in reactive oxygen species, generating reactions. The results showed that hesperidin has more potential iron chelation activities compared to deferoxamine, a popular chelator for treatment of chronic iron overload [70]. The antioxidant activity of hesperidin exhibits also by reducing the production of reactive oxygen species and increasing the activities of antioxidant enzymes, catalase and superoxide dismutase [71,72]. It should be mentioned that hesperidin exhibits lower antioxidant activity in comparison to its aglycone form, alike other flavonoids [60]. Citrus peels showed higher antioxidant ability than pulp due to its high content of flavonoids, vitamin C and carotenoids [73]. Al-Ashaal and El-Sheltawy reported that hesperidin from orange peel extract was moderately active as an antioxidant agent; its capacity reached 36% against free radical DPPH· in comparison to 100% obtained for vitamin C [66].

Cardiovascular protective effects of hesperidin are expressed in decreasing diastolic blood pressure, glucose levels and various lipid profile parameters, reducing platelet aggregation and increasing in the expression of antioxidative enzymes [74,75,76,77]. Some literature stated that hesperidin also exhibits cardioprotective effect against doxorubicin cardiotoxicity, which is widely used anticancer drug [14,74]. Razaee et al. found the protective effects of hesperidin against CO-induced cardiac injury in rat exposed to CO [78].

Administration of hesperidin decreased the expression of zinc finger E-box binding homeobox 2 (ZEB2, a transcription factor that binds to specific regions of DNA) by upregulating the expression of miRNA-132, which in turn promoted apoptosis and inhibited the proliferation of non-small cell lung cancer cells in mice [79]. Citrus peel extracts have been proven to be a promising therapeutic agent for diabetes mellitus, characterized by defects in insulin metabolism that can alter carbohydrate, protein and fat metabolism [2,4,80,81]. A review of the literature indicates that obesity is connected with insulin resistance and pancreatic β-cell dysfunction [82,83].

Mounting evidence has demonstrated that hesperidin possesses an inhibitory effect against the development of neurodegenerative conditions such as Alzheimer’s and Parkinson’s diseases [84,85,86,87]. It showed involvement of immunity in the development and progression of neurodegenerative disorders [88]. Hesperidin’s neuroprotective potential is mediated by the improvement of neural growth factors and endogenous antioxidant defense functions, diminishing neuro-inflammatory and apoptotic pathways. Dietary supplements containing hesperidin can significantly improve cerebral blood flow, cognition, and memory performance [84]. Several investigators have dedicated their effort to explore neuropharmacological mechanisms and the molecular target of citrus flavonoids, including hesperidin [89,90].

The potential anti-inflammatory effects of hesperidin for its possible therapeutic application against diverse pathologies have been evaluated [91,92,93,94]. Xiao et al. used it to effectively enhance chondrogenesis (a process that leads to the establishment of cartilage and bone formation) of human mesenchymal stem cells to facilitate cartilage tissue repair [92]. The results presented by Homayouni et al. suggest that hesperidin supplementation may have anti-inflammatory and antihypertensive effects in type 2 diabetes [93].

**Table 2 nutrients-14-02387-t002:** Examples of the main biological activities of hesperidin.

Biological Activities	Method	Hesperidin Dose	Results	Ref.
Antioxidative	Evaluation of marker enzymes and antioxidant status in blood, tissues, bronchoalveolar lavage cells and fluid after subcutaneous injection of nicotine	25° mg/kg	Protection against the lung damage caused by nicotine, which induces the lipid peroxidation	[72]
	Examination the iron chelation activity on the brain tissue of iron-overloaded mice	50 mg/kg per day (4 weeks)	Strong chelation of excessive iron from the serum and deposit iron	[70]
Prevention of cardiovascular diseases	Analysis of biochemical, histopathological, ultrastructural and immunohistochemical studies of rat heart after isoproterenol induced cardiac hypertrophy	200 mg/kg/ per day (4 weeks)	Improved hemodynamic and cardiac function parameters with a reduction in the levels of cardiac injury markers	[77]
	Evaluation of the effect of orange peel extract on streptozotocin-induced diabetic nephropathy	200 mg/kg for 4 weeks	Improved renal functions, significant prevention of the increase of creatinine, urea and blood urea nitrogen levels	[14]
Anti-inflammatory	Evaluation of the effects on neutrophil recruitment, edema, colon lesions and cytokines production in a pre-clinical model of ulcerative colitis induced by acetic acid in mice	100 mg/kg in saline by oral gavage	Reduction of inflammation, increase in colon antioxidant status, inhibition of proinflammatory cytokines	[94]
	Determination of blood pressure, serum antioxidant capacity, tumor necrosis factor alpha and inflammatory markers	500 mg/day (6 weeks)	Hesperidin has antihypertensive and anti-inflammatory effects in type 2 diabetes	[93]
Anticancer	The effect of hesperidin on the proliferation and apoptosis of non-small cell lung cancer in mice	60 mg/kg per day	Modulation of antioxidative enzymes induced apoptosis, suppression of cancer cell proliferation and invasiveness	[79]
Neurodegenerative properties	Evaluation of learning and memory impairment by radial arm maze, elevated plus maze and passive avoidance tests as well as oxidative stress and expression of pro and antiapoptotic markers	100 mg/kg	Prevention of the cognitive deficits, biochemical anomalies and apoptosis associated with neuro-degenerative diseases, including Alzheimer’s disease, induced by AlCl_3_ treatment.	[84]
	Investigation of the protective effect on behavior and neurochemical alterations, levels of ROS in an animal model of Parkinson’s disease induced by 6-hidroxidopamine	50 mg/kg for 28 days	Preventing memory impairment in the Morris water maze test and depressive-like behavior in the tail suspension test Hesperidin attenuates the induced reduction in glutathione peroxidase, catalase activity and total reactive antioxidant potential	[86]

ROS—reactive oxygen species.

The increasing antimicrobial resistance to synthetic antibiotics has attracted the interest of scientists in the direction of the use of naturally occurring compounds as effective antibacterial agents [95]. Several reports have demonstrated that hesperidin can also act against different pathogenic bacteria [77,96,97,98]. It can directly inhibit bacterial growth or act indirectly by modulating the expression of virulence factors, both of which reduce microbial pathogenicity. Hesperidin supplementation may be useful as a prophylactic agent against SARS-CoV-2 by blocking several mechanisms of viral infection and replication [99,100].

Hesperidin has also been associated with other, except those mentioned above, beneficial health effects, such as UV protection, wound healing and cutaneous functions [7] and radioprotective protection against ionizing radiation-induced damage [101]. Together with the flavone diosmin, under the trade name Daflon^®^, it decreases capillary fragility and is recommended for treating venous circulation disorders (swollen legs, pain, nocturnal cramps) and for treating symptoms due to acute hemorrhoidal attack [102]. Interested readers can find more specific information regarding biological activities of hesperidin as well as the results of the preclinical studies and clinical trials in the recent review papers [65,73,75,84,96].

## 6. Conclusions

Due to the variety of pharmacological activities in the human body, hesperidin is one of the most interesting and promising bioflavonoids. Citrus fruits and juices are widely consumed worldwide and are readily available dietary sources for its intake. Supplements containing hesperidin, alone or in combination with other citrus bioflavonoids, are commercially available. Moreover, processing of citrus by-products represents a rich source of hesperidin, owing to the large amount of peel produced. Their utilization can be used for the production of novel nutraceuticals or for the improvement of older ones.

However, it should be considered that the biological activities of hesperidin and its solubility and stability are dependent to a large degree on its transformation during the absorption and metabolism in the gastrointestinal track, which includes sulfonation and glucuronidation. For this reason, several methodologies have been proposed to improve its properties. The future development of novel delivery systems for hesperidin should also focus on the possible interactions between the food matrix and the flavanone. Checking the potential application of co-encapsulation of two or more bioactive ingredients to generate a synergistic effect could also be very interesting.

## Figures and Tables

**Figure 1 nutrients-14-02387-f001:**
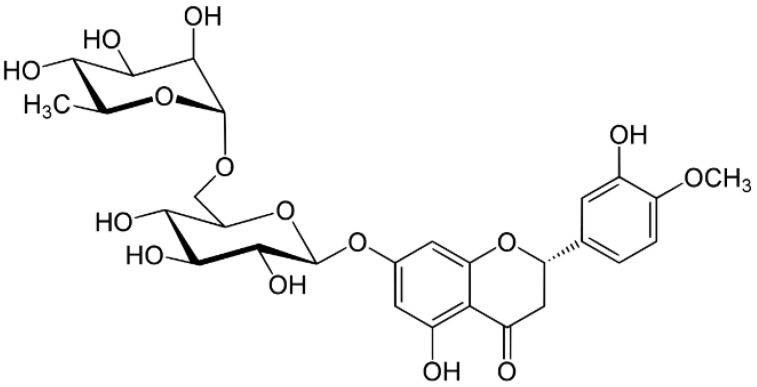
The chemical structure of hesperidin.

**Table 1 nutrients-14-02387-t001:** Examples of extraction conditions for isolation of hesperidin.

Sample	Conditions	Hesperidin mg/g dw	Ref.
Sweet orange pulp (*Citrus sinensis*)	HR, 55 °C, 20 min: 90% methanol 90% ethanol	24.77 17.93	[27]
Mandarin (*Citrus reticulata*) rinds	USE, 35 °C, 10 min: DMSO:methanol (1:1) 80% ethanol	32.0 5.46	[31]
Navel orange peels (*Citrus sinensis*)	40% ethanol, USE, 90 °C, 15 min	498	[23]
Thinned *Citrus unshiu* fruits	70% ethanol, MAE, 140 °C, 7 min DMSO:methanol (1:1), room temperature, 30 min	58.6 64.3	[29]
Peels of mandarin (*Citrus reticulata*)	70% methanol, PLE, 160 °C, 20 min 100% methanol, HR, 80 °C, 60 min	58.4 58.6	[35]
Peels of *Citrus unshi*	SWE, 160 °C, 10 min 70% methanol, 65 °C, 3 h 70% ethanol, 70 °C, 3 h water, 90 °C	73.0 22.4 37.3 3.1	[33]
*Sorbus tianschanica* leaves	1-Hexyl-3-methylimidazolium tetrafluoborate, MAE, 420 W, 19 min	0.48	[43]
Peels of mandarin (*Nobis tangerine*)	Choline chloride-acetamide, 45 °C, 25 min	38.0	[45]

HR—heat reflux; PLE—pressurized liquid extraction; MAE—microwave-assisted extraction; USE—ultrasound-assisted extraction; SWE—subcritical water extraction.

## Data Availability

Not applicable.

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
