# Peer review of "Hesperidin: A Review on Extraction Methods, Stability and Biological Activities"

_nutrients, 2022, doi:10.3390/nu14122387_

Round 1

Reviewer 1 Report

In this review, the author summarized the extraction and determination methods for quantification of hesperidin in natural and by-products, and discussed its chemical stability and biological activities. The review was well written and provided useful information.

Minor issue: I would suggest providing tables to list the methods with its advantage and limitation and the references so that it will be easier for the readers to get the information quickly.

Reviewer 2 Report

Dear Author,

The presented manuscript present short review on the extraction methods, chemical(?) stability and biological activities of hesperidin.There is great merit in preparation of review articles that provide compact up-to-date perspective to a specific compound, especially those of interdisciplinary interest. However, they should provide well organized and balanced information on the chosen topics. Provided manuscript needs substantial re-arrangement and more systematic presentation of the data. Some of the methodologies are given in detail and other topics are briefly mentioned without any discussion (see pdf file). Sustainability topic in utilization of by-products is scaterred and needs focus. Biological activities and bioavailability should by summarized in tables. Discussion on future direction in research and practice are missing. There is some potential in the paper, however, the author should upgrade the manuscript so to make it publishable and resubmit it after much needed corrections are made.

Round 2

Reviewer 2 Report

The corrections and additions fit well with the proposed changes.